# CAFE: Catastrophic Data Leakage in Vertical Federated Learning

**Xiao Jin**
Rensselaer Polytechnic Institute
jinx2@rpi.edu

**Pin-Yu Chen**
IBM Research
pin-yu.chen@ibm.com

**Chia-Yi Hsu**
National Yang Ming Chiao Tung University
chiayihsu8315@gmail.com

**Chia-Mu Yu**
National Yang Ming Chiao Tung University
chiamuyu@gmail.com

**Tianyi Chen**
Rensselaer Polytechnic Institute
chent18@rpi.edu

## Abstract

Recent studies show that private training data can be leaked through the gradients sharing mechanism deployed in distributed machine learning systems, such as federated learning (FL). Increasing batch size to complicate data recovery is often viewed as a promising defense strategy against data leakage. In this paper, we revisit this defense premise and propose an advanced data leakage attack with theoretical justification to efficiently recover batch data from the shared aggregated gradients. We name our proposed method as *catastrophic data leakage in vertical federated learning* (CAFE). Comparing to existing data leakage attacks, our extensive experimental results on vertical FL settings demonstrate the effectiveness of CAFE to perform large-batch data leakage attack with improved data recovery quality. We also propose a practical countermeasure to mitigate CAFE. Our results suggest that private data participated in standard FL, especially the vertical case, have a high risk of being leaked from the training gradients. Our analysis implies unprecedented and practical data leakage risks in those learning settings. The code of our work is available at https://github.com/DeRafael/CAFE.

## 1 Introduction

Federated learning (FL) [8, 24] is an emerging machine learning framework where a central server and multiple workers collaboratively train a machine learning model. Some existing FL methods consider the setting where each worker has data of a different set of subjects but sharing common features. This setting is also referred to data partitioned or horizontal FL (HFL). Unlike the HFL setting, in many learning scenarios, multiple workers handle data about the same set of subjects, but each has a different set of features. This case is common in finance and healthcare applications [6]. In these examples, data owners (e.g., financial institutions and hospitals) have different records of those users in their joint user base, and so, by combining their features through FL, they can establish a more accurate model. We refer to this setting as feature-partitioned or vertical FL (VFL).

Compared with existing distributed learning paradigms, FL raises new challenges including data heterogeneity and privacy [20]. To protect data privacy, only model parameters and the change of parameters (e.g., gradients) are exchanged between server and workers [19, 15]. Recent works have studied how a malicious worker can embed backdoors or replace the global model in FL [2, 3, 27]. Furthermore, as exchanging gradients is often viewed as privacy-preserving protocols, little attention has been paid to information leakage from public shared gradients and batch identities.

In the context of data security and AI ethics, the possibility of inferring private user data from the gradients in FL has received growing interests [10, 14, 21], known as the *data leakage* problems.

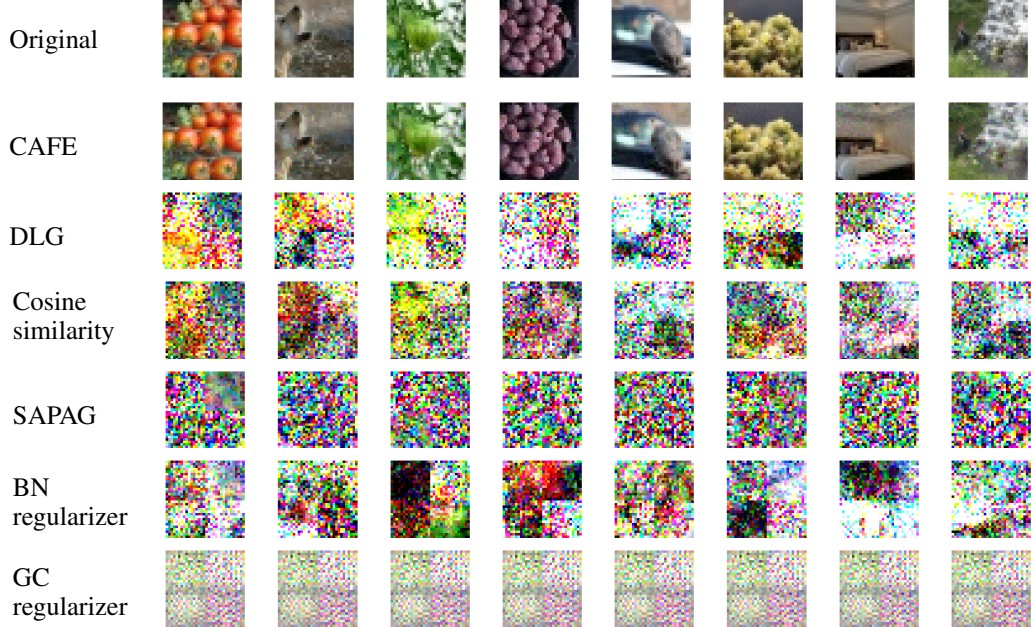

Figure 1: Visual comparison between CAFE (our method) with the state-of-the-art data leakage attacks including DLG [32], Cosine similarity [11], SAPAG [25], BN regularzier [29] and GC regularizer [29] on Linnaeus 5 in VFL (4 workers, batch size = 40 and batch ratio = 0.05).

Previous works have made exploratory efforts on data recovery through gradients. See Section 2 and Table 1 for details. However, existing approaches often have the limitation of scaling up large-batch data recovery and are lacking in theoretical justification on the capability of data recovery, which may give a false sense of security that increasing the data batch size during training can prevent data leakage [30]. Some recent works provide sufficient conditions for guaranteed data recovery, but the assumptions are overly restrictive and can be sometimes impractical, such as requiring the number of classes to be much larger than the number of recovered data samples [29].

To enhance scalability in data recovery and gain fundamental understanding on data leakage in VFL, in this paper we propose an advanced data leakage attack with theoretical analysis on the data recovery performance, which we call *catastrophic data leakage in vertical federated learning* (CAFE). As an illustration, Figure 1 demonstrates the effectiveness of CAFE for large-batch data recovery compared to existing methods. The contributions of this paper are summarized as follows.

**C1)** We develop a new data leakage attack named CAFE to overcome the limitation of current data leakage attacks on VFL. Leveraging the novel use of data index and internal representation alignments in VFL, CAFE is able to recover large-scale data in general VFL protocols.

**C2)** We provide theoretical guarantees on the recovery performance of CAFE, which permeates three steps of CAFE: (I) recovering gradients of loss with respect to the outputs of the first fully connected (FC) layer; (II) recovering inputs to the first FC layer; (III) recovering the original data.

**C3)** To mitigate the data leakage attack by CAFE, we develop a defense strategy which leverages the fake gradients and preserves the model training performance.

**C4)** We conduct extensive experiments on both static and dynamic VFL training settings to validate the superior data recovery performance of CAFE over state-of-the-art methods.

## 2 Related Work

Recovering private training data from gradients has gained growing interests in FL. Recently, a popular method termed deep leakage from gradients (DLG) [32] has been developed to infer training data in an efficient way without using any generative models or prior information. However, DLG

Table 1: Comparison of CAFE with state-of-the-art data leakage attack methods in FL.

| Method | Optimization terms | Reported maximal batch size | Training while attacking | Theoretical guarantee | Additional information other than gradients |
|---|---|---|---|---|---|
| DLG [32] | $\ell_2$ distance between real and fake gradients | 8 | No | No | No |
| iDLG [30] | $\ell_2$ distance | 8 | No | Yes | No |
| Inverting Gradients [11] | Cosine similarity, TV norm | 8 100 (Mostly unrecognizable) | Yes | Yes | Number of local updates |
| A Framework for Evaluating Gradient Leakage [26] | $\ell_2$ distance, label based regualrizer | 8 | No | Yes | No |
| SAPAG [25] | Gaussian kernel based funciton | 8 | No | No | No |
| R-GAP [31] | recursive gradient loss | 5 | No | Yes | The rank of matrix A defined in [31] |
| Theory oriented [22] | $\ell_2$ distance, $\ell_1$ distances of the recovered feature map | 32 | No | Yes | Number of Exclusive activated neurons |
| GradInversion[29] | Fidelity regularizers, Group consistency regularizers | 48 | No | No | Batch size $\ll$ number of classes & Non repeating labels in a batch |
| CAFE (ours) | $\ell_2$ distance, TV norm, Internal representation norm | 100 (our hardware limit) | Yes | Yes | Batch indices |

lacks generalizability on model architecture and weight distribution initialization [25]. In [30], an analytical approach has been developed to extract accurate labels from the gradients. In [11], another analytical approach has been developed to derive the inputs before a fully connected (FC) layer. However, in [11], their method only works on a single sample input and fails to extend on a batch of data. In [22], a new approach has been developed by recovering the batch inputs before the FC layer through solving linear equations. However, strong assumptions have been made for solving the equations and cannot guarantee data recovery in more general cases. In [9], it is claimed that a convolutional layer can always be converted to a FC layer. However, the gradients of the original convolutional layer are still different from the gradients of the converted FC layer, which impedes data recovery. Besides the new loss function proposed in [11], several previous works design new loss functions or regularizers based on DLG and try to make their algorithms work on more general models and weight distribution initialization. In [25], a new Gaussian kernel based gradient difference is used as the distance measure. In [31], a recursive method attack procedure has been developed to recover data from gradients. However, in both [25] and [31], the quality of recovery on batch data is degraded. A recent work [29] proposes an algorithm named GradInversion to reconstruct images from noise based on given gradients. However, their theory and algorithm are mostly built on strong assumptions and empirical observations. Although they successfully reconstruct a batch of training data, the reported batch size is still no larger than 48.

## 3 CAFE: Catastrophic Data Leakage in Vertical Federated Learning

In this section, we will introduce some necessary background of VFL and present our novel attack method. We consider the attack scenario where a honest-but-curious server follows the regular VFL protocols but intends to recover clients' private data based on the aggregated gradients. Our method is termed *CAFE: Catastrophic data leakage in vertical federated learning*. While CAFE can be applied to any type of data, without loss of generality, we use image datasets throughout the paper.

### 3.1 Preliminaries

**VFL setting.** FL can be categorized into horizontal and vertical FL settings [16]. In this paragraph, we provide necessary background of VFL. Consider a set of $M$ clients: $\mathcal{M} = \{1, 2, \ldots, M\}$. A dataset of $N$ samples $\mathcal{D} = \{(\mathbf{x}_n, y_n)\}_{n=1}^N$ are maintained by the $M$ local clients, where $n$ is the data index. Each client $m$ in $\mathcal{M}$ is associated with a unique features set. A certain data point $\mathbf{x}_n$ in $\mathcal{D}$ can be denoted by $\mathbf{x}_n = [\mathbf{x}_{n,1}^\top, \mathbf{x}_{n,2}^\top, \ldots, \mathbf{x}_{n,M}^\top]^\top$ where $\mathbf{x}_{n,m}$ is the $m$-th partition of the $n$-th sample

vector. The label set $\{y_n\}_{n=1}^N$ can be viewed as a special feature and is kept at the server or a certain local worker. Throughout this paper, we mainly study the VFL setting. CAFE can also be applied to HFL if the data indices of each randomly selected batch are known to workers during training.

**Use case of VFL.** VFL is suitable for cases where multiple data owners share the same data identity but their data differ in feature space. Use cases of VFL appear in finance, e-commerce, and health. For example, in medical industry, test results of the same patient from different medical institutions are required to diagnose whether the patient has a certain disease or not, but institutions tend not to share raw data. Figure 2 gives an example of VFL in medical industry.

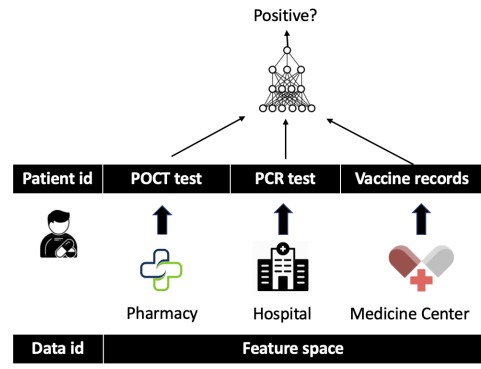

Figure 2: VFL among medical institutions

**Batch indices in each iteration.** For a given batch size $K$, we define a set of vectors with binary entries as $\mathcal{S} = \{\mathbf{s}_1, \mathbf{s}_2, \ldots, \mathbf{s}_i, \ldots\}$ with $|\mathcal{S}| = \binom{N}{K}$. For each vector $\mathbf{s}_i \in \mathbb{R}^N$ in $\mathcal{S}$, its $n$-th element $\mathbf{s}_i[n]$ can be either 0 or 1. There are in total $K$ enires of '1' in $\mathbf{s}_i$. In each iteration $t$, the server randomly selects one element from set $\mathcal{S}$ denoted by $\mathbf{s}^t$, where $\mathbf{s}^t[n]$ is the $n$th element in $\mathbf{s}^t$. The selected batch samples in the $t$-th iteration are denoted by $\mathcal{D}(\mathbf{s}^t) = \{(\mathbf{x}_n, y_n)|\mathbf{s}^t[n] = 1\}$.

**Loss function and gradients.** We assume that the model is a neural network parameterized by $\boldsymbol{\Theta}$, where the first FC layer is parameterized by $\boldsymbol{\Theta}_1 \in \mathbb{R}^{d_1 \times d_2}$ and its bias is $\mathbf{b}_1 \in \mathbb{R}^{d_2}$. The loss function on the batch data $\mathcal{D}(\mathbf{s}^t)$ and on the entire training data $\mathcal{D}$ is, respectively, denoted by

$$\mathcal{L}(\boldsymbol{\Theta}, \mathcal{D}(\mathbf{s}^t)) := \frac{1}{K}\sum_{n=1}^N \mathbf{s}^t[n]\mathcal{L}(\boldsymbol{\Theta}, \mathbf{x}_n, y_n) \quad \text{and} \quad \mathcal{L}(\boldsymbol{\Theta}, \mathcal{D}) := \frac{1}{N}\sum_{n=1}^N \mathcal{L}(\boldsymbol{\Theta}, \mathbf{x}_n, y_n). \quad (1)$$

The gradients of losses w.r.t. $\boldsymbol{\Theta}$ is denoted as

$$\nabla_{\boldsymbol{\Theta}} \mathcal{L}(\boldsymbol{\Theta}, \mathcal{D}(\mathbf{s}^t)) := \frac{\partial \mathcal{L}(\boldsymbol{\Theta}, \mathcal{D}(\mathbf{s}^t))}{\partial \boldsymbol{\Theta}} = \frac{1}{K}\sum_{n=1}^N \mathbf{s}^t[n]\frac{\partial \mathcal{L}(\boldsymbol{\Theta}, \mathbf{x}_n, y_n)}{\partial \boldsymbol{\Theta}}. \quad (2)$$

And similarly, we define $\nabla_{\boldsymbol{\Theta}} \mathcal{L}(\boldsymbol{\Theta}, \mathcal{D})$.

### 3.2 Why large-batch data leakage attack is difficult?

We motivate the design of our algorithm by providing some intuition on why performing large-batch data leakage from aggregated gradients is difficult [32]. Assume that $K$ images are selected as the inputs for a certain learning iteration. We define the selected batch data as $\mathcal{D}' = \{(\mathbf{x}_n, y_n)\}$. Likewise, the batched 'recovered data' is denoted by $\hat{\mathcal{D}}' = \{(\hat{\mathbf{x}}_n, \hat{y}_n)\}$. Then the objective function is

$$\hat{\mathcal{D}}' = \arg\min_{\hat{\mathcal{D}}'} \left\| \frac{1}{K}\sum_{(\mathbf{x}_n, y_n) \in \mathcal{D}} \nabla_{\boldsymbol{\Theta}}\mathcal{L}(\boldsymbol{\Theta}, \mathbf{x}_n, y_n) - \frac{1}{K}\sum_{(\hat{\mathbf{x}}_n, \hat{y}_n) \in \hat{\mathcal{D}}'} \nabla_{\boldsymbol{\Theta}}\mathcal{L}(\boldsymbol{\Theta}, \hat{\mathbf{x}}_n, \hat{y}_n) \right\|^2. \quad (3)$$

Note that in (3), the dimensions of the aggregated gradients is fixed. However, as $K$ increases, the cardinality of $\hat{\mathcal{D}}'$ and $\mathcal{D}'$ rise. When $K$ is sufficiently large, it will be more challenging to find the "right" solution $\hat{\mathcal{D}}'$ of (3) corresponding to the ground-truth dataset $\mathcal{D}'$. On the other hand, CAFE addresses this issue of large-batch data recovery by data index alignment (defined in next subsection), which can effectively exclude undesired solutions. We discuss a specific example in Appendix B.

### 3.3 CAFE implementation

The main idea of our algorithm is that we divide the entire data leakage attack procedure into several steps. Specifically, we fully recover the inputs to the first FC layers of the model that we term the internal representation with theoretical guarantee and use the internal representation as a learnt regularizer to help improve the performance of data leakage attack. During the process, to overcome

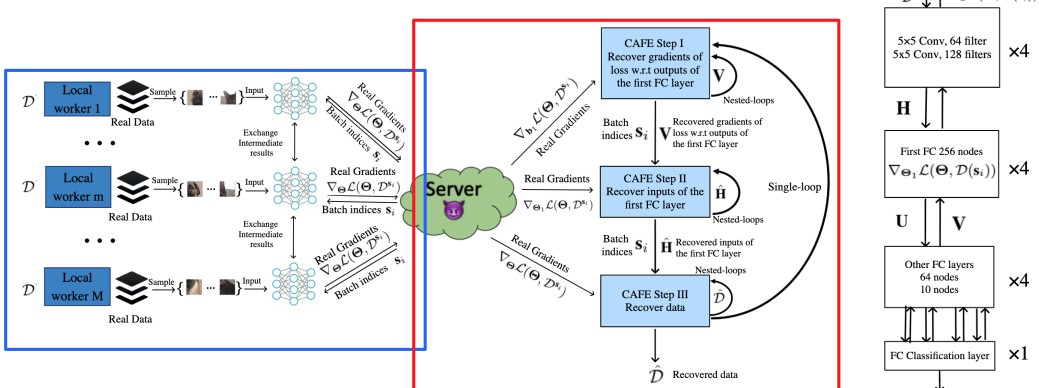

Figure 3: Overview of CAFE. The left part (blue box) performs the regular VFL protocol and the right part (red box) illustrates the main steps of CAFE.

Figure 4: Model structure in VFL.

the difficulty mentioned in Section 3.2, we fully use the batch data index known by the attacker in the VFL setting so that the system equation in (3) can be determined instead of undetermined.

**Prerequisite:** Notably, CAFE can be readily applied to existing VFL protocols where the batch data indices is assigned or other deep learning protocols as long as the batch data indices are given. In Figure 3, the blue box represents the VFL paradigm and the red box denotes the attack paradigm.

In a typical VFL process, the server sends public key to local workers and decides the data indices in each iteration of training and evaluation [7, 23]. During the training process, local workers exchange their intermediate results with others to compute gradients and upload them. Therefore, the server has access to *both* of the model parameters and their gradients. Since data are vertically partitioned among different workers, for each batch, the server (acting as the attacker) needs to send a data index or data id list to all the local workers to ensure that data with the same id sequence have been selected by each worker [28] and we name this step as *data index alignment*. *Data index alignment* turns out to be an inevitable step in the vertical training process, which provides the server (the attacker) an opportunity to control the selected batch data indices.

In the rest of this subsection, we explain our algorithm CAFE in detail, which consists of three steps.

**Step I: Recover the gradients of loss w.r.t the outputs of the first FC layer.** As shown in Figure 4, for a certain data point $\mathbf{x}_n$, we denote the inputs to the first FC layer as $\mathbf{h}_n = h(\mathbf{\Theta}_c, \mathbf{x}_n) \in \mathbb{R}^{d_1}$ where $h$ is the forward function and $\mathbf{\Theta}_c$ is the parameters before the first FC layer. Let $\mathbf{u}_n$ denote the outputs of the first FC layer in the neural network, given by

$$\mathbf{u}_n = \mathbf{\Theta}_1^\top \mathbf{h}_n + \mathbf{b}_1 \in \mathbb{R}^{d_2}. \tag{4}$$

For the training data $\mathcal{D}$, the corresponding inputs before the first FC layer are concatenated as $\mathbf{H} = [\mathbf{h}_1, \mathbf{h}_2, \ldots, \mathbf{h}_N]^\top \in \mathbb{R}^{N \times d_1}$ and the corresponding outputs of the first FC layer are concatenated as $\mathbf{U} = [\mathbf{u}_1, \mathbf{u}_2, \ldots, \mathbf{u}_N]^\top \in \mathbb{R}^{N \times d_2}$. The gradients of loss w.r.t $\mathbf{U}$ can be denoted by

$$\nabla_{\mathbf{U}}\mathcal{L}(\mathbf{\Theta}, \mathcal{D}) = \frac{1}{N}[\nabla_{\mathbf{u}_1}\mathcal{L}(\mathbf{\Theta}, \mathbf{x}_1, y_1), \nabla_{\mathbf{u}_2}\mathcal{L}(\mathbf{\Theta}, \mathbf{x}_2, y_2), \ldots, \nabla_{\mathbf{u}_N}\mathcal{L}(\mathbf{\Theta}, \mathbf{x}_N, y_N)]^\top$$

$$= \frac{1}{N}\left[\frac{\partial \mathcal{L}(\mathbf{\Theta}, \mathbf{x}_1, y_1)}{\partial \mathbf{u}_1}, \frac{\partial \mathcal{L}(\mathbf{\Theta}, \mathbf{x}_2, y_2)}{\partial \mathbf{u}_2}, \ldots, \frac{\partial \mathcal{L}(\mathbf{\Theta}, \mathbf{x}_N, y_N)}{\partial \mathbf{u}_N}\right]^\top \in \mathbb{R}^{N \times d_2}. \tag{5}$$

For a batch of data in the $t$-th iteration $\mathcal{D}(\mathbf{s}^t)$, we have

$$\nabla_{\mathbf{b}_1}\mathcal{L}(\mathbf{\Theta}, \mathcal{D}(\mathbf{s}^t)) = \frac{1}{K}\sum_{n=1}^{N}\mathbf{s}^t[n]\frac{\partial \mathcal{L}(\mathbf{\Theta}, \mathbf{x}_n, y_n)}{\partial \mathbf{b}_1} = \sum_{n=1}^{N}\mathbf{s}^t[n]\frac{1}{K}\sum_{z=1}^{N}\mathbf{s}^t[z]\frac{\partial \mathcal{L}(\mathbf{\Theta}, \mathbf{x}_z, y_z)}{\partial \mathbf{u}_n}$$

$$= \sum_{n=1}^{N}\mathbf{s}^t[n]\nabla_{\mathbf{u}_n}\mathcal{L}(\mathbf{\Theta}, \mathcal{D}(\mathbf{s}^t)). \tag{6}$$

Although we do not have access to $\nabla_{\mathbf{U}}\mathcal{L}(\mathbf{\Theta}, \mathcal{D})$ as gradients are only given w.r.t. the model parameters, we can successfully recover it through an iterative optimization process.

**Algorithm 1** Recover the gradients $\nabla_{\mathbf{U}}\mathcal{L}(\Theta, \mathcal{D})$ ( regular VFL and attacker )

1: Given model parameters $\Theta$ and $\mathbf{V} \sim \mathcal{U}^{N \times d_1}$
2: **for** $t = 1, 2, \ldots, T$ **do**
3:     Server select $\mathbf{s}^t$ from $\mathcal{S}$
4:     Server broadcasts $\Theta$ and $\mathbf{s}^t$ to all workers
5:     **for** $m = 1, 2, \ldots, M$ **do**
6:        Worker $m$ takes real batch data
7:        Worker $m$ exchanges intermediate results with other workers and computes $\nabla_{\Theta}\mathcal{L}(\Theta, \mathcal{D}(\mathbf{s}^t))$
8:        Worker $m$ uploads $\nabla_{\Theta}\mathcal{L}(\Theta, \mathcal{D}(\mathbf{s}^t))$
9:     **end for**
10:    Server computes $\nabla_{\mathbf{b}_1}\mathcal{L}(\Theta, \mathcal{D}(\mathbf{s}^t))$
11:    Server computes $\mathcal{F}_1(\mathbf{V}; \mathbf{s}^t)$ in (7)
12:    Server updates $\mathbf{V}$ with $\nabla_{\mathbf{V}}\mathcal{F}_1(\mathbf{V}; \mathbf{s}^t)$
13: **end for**

**Algorithm 2** Recover the inputs to the first FC layer $\mathbf{H}$ ( regular VFL and attacker )

1: Given $\Theta$, trained $\mathbf{V}$, initialize $\hat{\mathbf{H}} \sim \mathcal{U}^{N \times d_2}$
2: **for** $t = 1, 2, \ldots, T$ **do**
3:     Server select $\mathbf{s}^t$ from $\mathcal{S}$.
4:     Server broadcasts $\Theta$ and $\mathbf{s}^t$ to all workers
5:     **for** $m = 1, 2, \ldots, M$ **do**
6:        Worker $m$ takes real batch data
7:        Worker $m$ exchanges intermediate results with other workers and computes $\nabla_{\Theta}\mathcal{L}(\Theta, \mathcal{D}(\mathbf{s}^t))$
8:        Worker $m$ uploads $\nabla_{\Theta}\mathcal{L}(\Theta, \mathcal{D}(\mathbf{s}^t))$
9:     **end for**
10:    Server computes $\nabla_{\Theta_1}\mathcal{L}(\Theta, \mathcal{D}(\mathbf{s}^t))$
11:    Server computes $\mathcal{F}_2(\hat{\mathbf{H}}; \mathbf{s}^t)$ in (8)
12:    Server updates $\hat{\mathbf{H}}$ with $\nabla_{\hat{\mathbf{H}}}\mathcal{F}_2(\hat{\mathbf{H}}; \mathbf{s}^t)$
13: **end for**

**Algorithm 3** CAFE (Nested-loops)

1: Given model parameters $\Theta$, initialize $\mathbf{V} \sim \mathcal{U}^{N \times d_1}, \hat{\mathbf{H}} \sim \mathcal{U}^{N \times d_2}, \hat{\mathcal{D}} = \{\hat{\mathbf{x}}_n, \hat{y}_n\}_{n=1}^N$
2: Run Algorithms 1 and 2 each for $T$ iterations
3: **for** $t = 1, 2, \ldots, T$ **do**
4:     Run Step 3-10 in Algorithm 1 once
5:     Server computes $\nabla_{\Theta}\mathcal{L}(\Theta, \mathcal{D}(\mathbf{s}^t))$
6:     Server computes the fake global aggregated gradients $\nabla_{\Theta}\mathcal{L}(\Theta, \hat{\mathcal{D}}(t))$
7:     Server computes CAFE loss $\mathcal{F}_3(\hat{\mathcal{D}}; \mathbf{s}^t)$ in (9)
8:     Server updates $\hat{\mathcal{D}}$ with $\nabla_{\hat{\mathcal{D}}}\mathcal{F}_3(\hat{\mathcal{D}}; \mathbf{s}^t)$
9: **end for**

**Algorithm 4** CAFE (Single-loop)

1: Given model parameters $\Theta$, initialize $\mathbf{V} \sim \mathcal{U}^{N \times d_1}, \hat{\mathbf{H}} \sim \mathcal{U}^{N \times d_2}, \hat{\mathcal{D}} = \{\hat{\mathbf{x}}_n, \hat{y}_n\}_{n=1}^N$
2: **for** $t = 1, 2, \ldots, T$ **do**
3:     Run Step 3-10 in Algorithm 1 once
4:     Server computes $\nabla_{\Theta}\mathcal{L}(\Theta, \mathcal{D}(\mathbf{s}^t))$ including $\nabla_{\mathbf{b}_1}\mathcal{L}(\Theta, \mathcal{D}(\mathbf{s}^t)), \nabla_{\Theta_1}\mathcal{L}(\Theta, \mathcal{D}(\mathbf{s}^t))$
5:     Run Step 11 - 12 in Algorithm 1 once
6:     Run Step 11 - 12 in Algorithm 2 once
7:     Server computes CAFE loss $\mathcal{F}_3(\hat{\mathcal{D}}; \mathbf{s}^t)$ in (9)
8:     Server updates $\hat{\mathcal{D}}$ with $\nabla_{\hat{\mathcal{D}}}\mathcal{F}_3(\hat{\mathcal{D}}; \mathbf{s}^t)$
9: **end for**

Specifically, we randomly initialize an estimate of $\nabla_{\mathbf{U}}\mathcal{L}(\Theta, \mathcal{D})$ denoted as $\mathbf{V}$, e.g., $\mathbf{V} = [\mathbf{v}_1, \mathbf{v}_2, \ldots, \mathbf{v}_n, \ldots, \mathbf{v}_N]^\top \in \mathbb{R}^{N \times d_2}$, where $\mathbf{v}_n = [v_{n,1}, v_{n,1}, \ldots, v_{n,d_2}]^\top \in \mathbb{R}^{d_2}$. Given $\nabla_{\mathbf{b}_1}\mathcal{L}(\Theta, \mathcal{D}(\mathbf{s}^t))$, we recover $\nabla_{\mathbf{U}}\mathcal{L}(\Theta, \mathcal{D})$ by minimizing the following objective function

$$\mathbf{V}^* = \arg\min_{\mathbf{V}} \underbrace{\mathbb{E}_{\mathbf{s}_i \sim \mathbf{Unif}(\mathcal{S})}\left[\mathcal{F}_1(\mathbf{V}; \mathbf{s}_i)\right]}_{:=\mathcal{F}_1(\mathbf{V})} \quad \text{with} \quad \mathcal{F}_1(\mathbf{V}; \mathbf{s}_i) := \left\|\mathbf{V}^\top \mathbf{s}_i - \nabla_{\mathbf{b}_1}\mathcal{L}(\Theta, \mathcal{D}(\mathbf{s}_i))\right\|_2^2. \quad (7)$$

In each iteration $t$, the objective function of Step I is given by $\mathcal{F}_1(\mathbf{V}; \mathbf{s}^t)$.

The first step of CAFE is summarized in Algorithm 1, which enjoys the following guarantee.

**Theorem 1.** If $K < N$, the objective function $\mathcal{F}_1(\mathbf{V})$ in (7) is strongly convex in $\mathbf{V}$. For a fixed $\Theta$, applying SGD to (7) guarantees the convergence to the ground truth almost surely.

When the batch size $K$ is smaller than the number of total data samples $N$, the Hessian matrix of $\mathcal{F}_1(\mathbf{V})$ is shown to be strongly convex in Appendix C and the convergence is guaranteed according to [23]. Step I is essential in CAFE because we separate the gradients of loss w.r.t each single input to the first FC layer from the aggregated gradients in this step.

**Step II: Recover inputs to the first FC layer.** Using the chain rule, we have $\nabla_{\Theta_1}\mathcal{L}(\Theta, \mathcal{D}) = \mathbf{H}^\top \nabla_{\mathbf{U}}\mathcal{L}(\Theta, \mathcal{D}) \in \mathbb{R}^{d_1 \times d_2}$. We randomly initialize an estimate of $\mathbf{H}$ as $\hat{\mathbf{H}} = [\hat{\mathbf{h}}_1, \hat{\mathbf{h}}_2, \ldots, \hat{\mathbf{h}}_n, \ldots, \hat{\mathbf{h}}_N]^\top \in \mathbb{R}^{N \times d_1}$, where $\hat{\mathbf{h}}_n = [\hat{h}_{n,1}, \hat{h}_{n,1}, \ldots, \hat{h}_{n,d_1}]^\top \in \mathbb{R}^{d_1}$. Given $\nabla_{\Theta_1}\mathcal{L}(\Theta, \mathcal{D}(\mathbf{s}^t))$ and $\mathbf{V}^*$, we recover $\mathbf{H}$ by minimizing the following objective

$$\hat{\mathbf{H}}^* = \arg\min_{\hat{\mathbf{H}}} \underbrace{\mathbb{E}_{\mathbf{s}_i \sim \mathbf{Unif}(\mathcal{S})}\mathcal{F}_2(\hat{\mathbf{H}}; \mathbf{s}_i)}_{:=\mathcal{F}_2(\hat{\mathbf{H}})} \quad \text{with} \quad \mathcal{F}_2(\hat{\mathbf{H}}; \mathbf{s}_i) := \left\|\sum_{n=1}^N \mathbf{s}_i[n]\hat{\mathbf{h}}_n(\mathbf{v}_n^*)^\top - \nabla_{\Theta_1}\mathcal{L}(\Theta, \mathcal{D}(\mathbf{s}_i))\right\|_F^2.$$

$$(8)$$

In each iteration $t$, the objective function of Step II can be denoted by $\mathcal{F}_2(\hat{\mathbf{H}}; \mathbf{s}^t)$.

Through the first two steps, parts of the information about the data have already been leaked. Step II also has the following guarantee.

**Theorem 2.** If $N < d_2$ and $\mathrm{Rank}(\mathbf{V}^*) = N$, the objective function $\mathcal{F}_2(\hat{\mathbf{H}})$ is strongly convex. When $\Theta$ keeps unchanged, applying SGD guarantees the convergence of $\hat{\mathbf{H}}$ to $\mathbf{H}$.

Our experiment setting satisfies the assumption, e.g., $N = 800$ and $d_2 = 1024$, and thus the convergence is guaranteed according to [23]. The proof of Theorem 2 can be found in Appendix D. In some simple models such as *logistic regression* or neural network models *only containing FC layers*, the attack will recover the data only by implementing the first two steps.

**Step III: Recover data.** We randomly initialize the fake data and fake labels followed by uniform distribution $\hat{\mathcal{D}} = \{\hat{\mathbf{x}}_n, \hat{y}_n\}_{n=1}^N$. According to equation (4), we have $\widetilde{\mathbf{h}}_n = h(\Theta_c, \hat{\mathbf{x}}_n) \in \mathbb{R}^{d_1}$.

Given $\nabla_{\Theta}\mathcal{L}(\Theta, \mathcal{D}(\mathbf{s}_i))$ and $\hat{\mathbf{H}}^*$, our objective function in the last step is

$$\hat{\mathcal{D}}^* = \arg\min_{\hat{\mathcal{D}}} \mathbb{E}_{\mathbf{s}_i \sim \mathbf{Unif}(\mathcal{S})}[\mathcal{F}_3(\hat{\mathcal{D}}; \mathbf{s}_i)] \tag{9}$$

$$\text{with } \mathcal{F}_3(\hat{\mathcal{D}}; \mathbf{s}_i) := \alpha\left\|\nabla_{\Theta}\mathcal{L}(\Theta, \mathcal{D}(\mathbf{s}_i)) - \nabla_{\Theta}\mathcal{L}(\Theta, \hat{\mathcal{D}}(\mathbf{s}_i))\right\|_2^2 + \beta\underline{\mathrm{TV}}_\xi(\hat{\mathcal{X}}(\mathbf{s}_i)) + \gamma\sum_{n=1}^N \left\|\mathbf{s}_i[n](\hat{\mathbf{H}}_n^* - \widetilde{\mathbf{h}}_n)\right\|_2^2$$

where $\alpha, \beta$ and $\gamma$ are coefficients, $\underline{\mathrm{TV}}_\xi(\hat{\mathcal{X}}(\mathbf{s}_i))$ is the truncated total variation (TV) norm which is 0 if the TV-norm of $\hat{\mathcal{X}}(\mathbf{s}_i) = \{\hat{\mathbf{x}}_n | \mathbf{s}_i[n] = 1\}$ is smaller than $\xi$, and $\hat{\mathcal{D}}(\mathbf{s}_i) = \{\{\hat{\mathbf{x}}_n, \hat{y}_n\} | \mathbf{s}_i[n] = 1\}$. In each iteration $t$, the objective function of step III is $\mathcal{F}_3(\hat{\mathcal{D}}; \mathbf{s}^t)$. The first term in (9) is the $\ell_2$ norm in [32]. The second term is the TV norm and the last term is the internal representation norm regularizer. We also define $\nabla_{\hat{\mathcal{D}}}\mathcal{F}_3(\hat{\mathcal{D}}; \mathbf{s}^t) = \{\nabla_{\hat{\mathbf{x}}_n}\mathcal{F}_3(\hat{\mathcal{D}}; \mathbf{s}^t), \nabla_{\hat{y}_n}\mathcal{F}_3(\hat{\mathcal{D}}; \mathbf{s}^t)\}_{n=1}^N$.

To ensure attacking efficiency, we consider two flexible update protocols in CAFE — Algorithm 3: CAFE (Nested-loops) and Algorithm 4: CAFE (Single-loop). Empirically, Algorithm 4 will take fewer iterations than those of Algorithm 3. More details can be found in the experiment results in Section 4.2. We also discuss the theoretical guarantee for each step and its proof in Appendix E.

### 3.4 Defense strategy: Leveraging fake gradients as a countermeasure to CAFE

Although CAFE comes with theoretical recovery guarantees, the underlying premise is that the clients will upload true (correct) gradients for aggregation. Therefore, we propose an intuitive and practical approach to mitigate CAFE by requiring each client to upload fake (but similar) gradients, resulting in incorrect data recovery via CAFE. Specifically, to solve the problem of leakage from true gradients, we design a defense called *Fake Gradients* and summarize it in Algorithm 5 of Appendix F. The main idea of this defense is that attackers will aim to match wrong gradients and invert incorrect inputs to the first FC layer so that attackers cannot recover the true training data. The defending strategy in Algorithm 5 (Appendix F) can be added between Line 8 and 9 in Algorithms 1 and 2.

As summarized in Algorithm 5 (Appendix F), each local worker can randomly generate gradients with the normal distribution $\mathcal{N}(0, \sigma^2)$ and sort the elements in descending order (Line 1, 2). At the same time, local workers also sort their true gradients in descending order and record indexes of the sorted items (Line 7). Then, one computes the $L_2$-norm distance between a true gradient and all fake gradients to find the nearest fake gradient (Line 12). Afterwards, we pair fake gradients to match true gradients by the sorted order (Line 17). This an important step so that we can keep large/small values at the same positions of true gradients. Finally, local workers upload the fake gradients to the server.

**Impact on model training.** Chen et al. [5] has proved that if the distance between the actual gradients and the gradient surrogate is smaller than a decreasing threshold, using the gradient surrogate to update the model still guarantees convergence. Building upon the results in [5], we set a sufficient threshold such that the distance between the fake gradients and the true gradients are smaller than the threshold. In this case, we can still achieve the learning performance as if true gradients are used.

Table 2: Comparison with the state-of-the-art ($M = 4$, $K = 40$, batch ratio = 0.05)

| PSNR Dataset / Method | CIFAR-10 | MNIST | Linnaeus 5 |
|---|---|---|---|
| CAFE | 31.83 | 43.15 | 33.22 |
| DLG | 9.29 | 7.96 | 7.14 |
| Cosine Similarity | 7.38 | 7.84 | 8.31 |
| SAPAG | 6.07 | 3.86 | 6.74 |
| BN regularizer | 18.94 | 13.38 | 8.09 |
| GC regularizer | 13.63 | 9.24 | 12.32 |

Table 3: PSNR vs batch size $K$ (800 data samples in total)

| PSNR Dataset / $K$ | CIFAR-10 | MNIST | Linnaeus 5 |
|---|---|---|---|
| 10 | 30.83 | 32.60 | 28.00 |
| 20 | 35.70 | 39.00 | 30.53 |
| 40 | 31.83 | 43.15 | 33.22 |
| 80 | 36.87 | 47.05 | 30.43 |
| 100 | 38.94 | 47.50 | 29.18 |

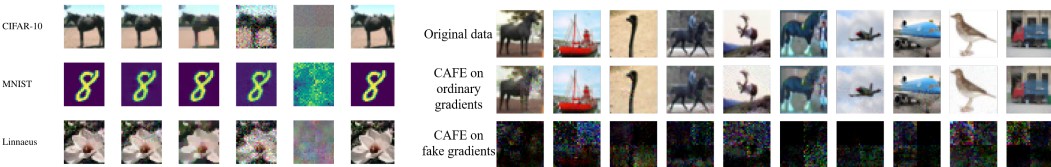

Figure 5: Visual comparison on the effect of auxiliary regularizers.

Figure 6: Visual comparison of the real and recovered data using ordinary and fake gradients.

# 4 Experiments

We conduct experiments on MNIST [18], CIFAR-10 [17] and Linnaeus 5 [4] datasets in VFL settings. The hyper-parameter settings are shown in Appendix G.1. Our algorithm recovers all the data participating in VFL with a relative large batch size (more than 40). Scaling up to our hardware limits (RTX 2080 and TITAN V), CAFE can leak as many as 800 images in the VFL setting including 4 workers with a batch size as large as 100. The neural network model architecture used in the simulation is shown in Figure 4. To measure the data leakage performance, we use the peak signal-to-noise ratio (PSNR) value and the mean squared error (MSE). Higher PSNR value of leaked data represents better performance of data recovery.

## 4.1 Comparison with the state-of-the-art

We compare CAFE with five state-of-the-art methods using the batch size of 40 images in each iteration. For fair comparisons, all methods were run on the the same model and iterations.

**i) DLG [32]:** The deep gradients leakage method is equivalent to replacing the objective function in (9) with the squared $\ell_2$ norm distance.

**ii) Cosine Similarity [11]:** The objective function is equivalent to replacing the objective function in (9) with the linear combination of cosine similarity and TV norm of the recovered images.

**iii) SAPAG [25]:** The objective function is equivalent to replacing the objective function in (9) with the Gaussian kernel based function.

**iv) Batch normalization (BN) regularizer [29]:** The objective function is equivalent to replacing the TV norm and internal representation norm in (9) with the batch normalization regularizer [29].

**v) Group consistency (GC) regularizer [29]:** The objective function is equivalent to replacing the TV norm and internal representation norm in (9) with the group consistency regularizer [29].

In GradInversion [29], several additional assumptions have been made. For example, the assumption of non-repeating labels in the batch is hard to be satisfied in datasets such as CIFAR-10, MNIST and Linnaeus 5. In those datasets, we use batch size of more than 40, which is larger than the number of classes (10 or 5). Nevertheless, we still compared our CAFE to the methods by using the batch normalization regularizer and group consistency regularizer mentioned in [29] in CAFE.

Theory-driven label inference methods have been proposed in [30] and [26]. However, our attack mainly deals with training data leakage rather than labels. In [22], the authors proposed a sufficient requirement that "each data sample has at least two exclusively activated neurons at the last but one layer". However, in our training protocol, the batch size is too large and it is almost impossible to ensure that each selected sample has at least two exclusively activated neurons. In [31], it is assumed that the method will only return a linear combination of the selected training data, which is a very restricted assumption. As the results, we did not compare to those methods in Table 2.

Table 4: Effect of auxiliary regularizers
($M = 4$, $K = 40$, batch ratio = 0.05)

| PSNR / Algorithm \ Datasets | CIFAR-10 | Linnaeus 5 | MNIST |
|---|---|---|---|
| CAFE | 31.83 | 33.22 | 43.15 |
| CAFE ($\alpha = 0$) | 33.93 | 28.62 | 31.93 |
| CAFE ($\xi = 0$) | 25.57 | 25.29 | 34.51 |
| CAFE ($\beta = 0$) | 18.25 | 23.22 | 31.98 |
| CAFE ($\gamma = 0$) | 12.51 | 12.37 | 6.34 |

Table 5: Nested-loops vs single-loop CAFE
($M = 4$, $K = 40$, batch ratio = 0.05)

| Iterations \ mode / Datasets | CIFAR-10 | MNIST | Linnaues 5 |
|---|---|---|---|
| Single loop | 7300 (8000) | 6600 (8000) | 12400 (20000) |
| Nested-loops Step I | 8000 (8000) | 8000 (8000) | 12428 (20000) |
| Nested-loops Step II | 2404 (8000) | 8000 (8000) | 20000 (20000) |
| Nested-loops Step III | 1635 (8000) | 2468 (8000) | 20000 (20000) |

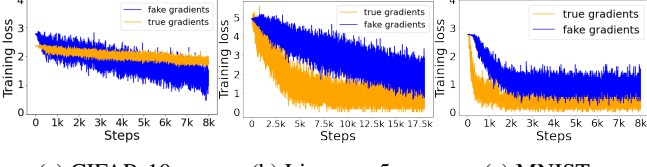

(a) CIFAR-10    (b) Linnaeus 5    (c) MNIST

Figure 7: Training loss of true gradients and fake gradients on CIFAR-10, Linnaeus 5 and MNIST.

Table 6: Effects of number of workers $M$
($K = 40$, batch ratio = 0.05)

| PSNR / $M$ \ Datasets | CIFAR-10 | Linnaeus 5 | MNIST |
|---|---|---|---|
| 4 | 31.83 | 33.22 | 43.15 |
| 16 | 28.39 | 39.85 | 39.28 |

CAFE outperforms these methods both qualitatively (Figure 1) and quantitatively (Table 2). Its PSNR values are always above 30 at the end of each CAFE attacking process, suggesting high data recovery quality. However, the PSNR of other methods are below 10 on all the three datasets.

## 4.2 Ablation study

We test CAFE under different batch size, network structure and with/without auxiliary regularizers.

**(i) PSNR via Batch size $K$.** Table 3 shows that the PSNR values always keep above 30 on CIFAR-10, above 32 on MNIST and above 28 on Linnaeus 5 when the batch size $K$ increases with fixed number of workers and number of total data points. The result implies that the increasing $K$ has almost no influence on data leakage performance of CAFE and it fails to be an effective defense.

**(ii) PSNR via Epoch.** Theoretically, given infinite number of iterations, we prove that we can recover $\nabla_{\mathbf{U}}\mathcal{L}$ and $\mathbf{H}$ because the respective objective function in (7) and (8) in our paper is strongly convex as long as $N < d_2$ and $\text{Rank}(\mathbf{V}^*) = N$ in Sections C and D of supplementary material. The corresponding experimental results and analysis are shown in Appendix G.2.

**(iii) Effect of regularizers.** Table 4 demonstrates the impact of regularizers. From Figure 5, adjusting the threshold $\xi$ prevents images from being over blurred during the reconstruction process. TV norm can eliminate the noisy patterns on the recovered images and increase the PSNR. We also find that the last term in (9), the internal representation norm regularizer, contributes most to the data recovery. In Table 4, CAFE still performs well without the first term ($\alpha = 0$) in (9). The reason is that the internal representation regularizer already allows data to be fully recovered. Notably, CAFE also performs well on MNIST even without the second term ($\beta = 0$) in (9). It is mainly due to that MNIST is a simple dataset that CAFE can successfully recover even without the TV-norm regularizer.

**(iv) Nested-loops vs single-loop.** We compare both modes of CAFE (Algorithms 3 and 4) on all datasets. In Table 5, the number of iterations is the maximum iterations at each step. For the CAFE (single-loop), if the objective function in step I (7) decreases below $10^{-9}$, we switch to step II. If the objective function in step II (8) decreases below $5 \times 10^{-9}$, we switch to step III. When the PSNR value reaches 27 on CIFAR-10, 30 on Linnaeus 5, 38 on MNIST, we stop both algorithms and record the iteration numbers. As shown in Table 5, CAFE single-loop requires fewer number of iterations. Meanwhile, it is difficult to set the loop stopping conditions in the CAFE Nested-loops mode. In particular, $\mathbf{V}^*$ and $\hat{\mathbf{H}}^*$ with low recovery precision may impact the data recovery performance.

**(v) Effects of number of workers $M$.** Although data are partitioned on feature space across workers, the dimension of the entire data feature space is fixed and independent of $M$. Therefore, increasing number of workers theoretically does not change the dimension of variables associated with data recovery in (3). In practice, different from HFL, where there could be hundreds of workers, in VFL,

Table 7: Attacking while training in VFL

| PSNR(lr) Setting / Dataset | 1 | 2 | 3 |
|---|---|---|---|
| CIFAR10 | 31.24 $(10^{-4})$ | 27.62 $(5 \times 10^{-4})$ | 25.22 $(10^{-3})$ |
| MNIST | 31.82 $(10^{-4})$ | 28.42 $(5 \times 10^{-4})$ | 23.60 $(10^{-3})$ |
| Linnaeus 5 | 30.74 $(10^{-6})$ | 21.45 $(5 \times 10^{-5})$ | 20.68 $(10^{-4})$ |

Table 8: Training while attacking on MNIST

| # of iterations | PSNR value | Training loss | Testing accuracy |
|---|---|---|---|
| 0 | 5.07 | 2.36 | 0.11 |
| 2000 | 11.68 | 2.31 | 0.27 |
| 6000 | 18.07 | 1.99 | 0.54 |
| 10000 | 18.12 | 1.82 | 0.64 |
| 15000 | 16.86 | 1.63 | 0.65 |
| 20000 | 20.72 | 1.68 | 0.68 |

the workers are typically financial organizations or companies. Therefore, the number of workers is usually small [13]. In Table 6, we compare the results of 4 workers with 16 workers following the same experiment setup. The CAFE performances are comparable.

### 4.3 Tests for attacking while training scenarios

Previous works have shown that DLG performs better on an untrained model than a trained one [11]. This is also true for CAFE. Our theoretical analysis can provide the partial reason. When the model is trained or even convergent, the real gradients of loss can be very small. It is possible that the value of the recovered $\nabla_{\mathbf{U}}\mathcal{L}(\Theta, \mathcal{D})$ will also be close to 0. In that case, it can be difficult to recover $\mathbf{H}$.

We also implement CAFE in the 'attacking while training' scenario, in which we continuously run the VFL process. When the model is training, both of the selected batch data and the model parameters change every iteration, which may cause the attack loss to diverge. However, from our experimental results in Table 7, CAFE is able to recover training images when the learning rate (lr) is relatively small. Increasing the learning rate renders data leakage more difficult because the model is making more sizeable parameter changes in each iteration, which can be regarded as an effective defense strategy. According to our experiment in Table 8, the model indeed converges with a relative small learning rate (e.g., Adam with learning rate $10^{-6}$, trained on 800 images, tested on 100 images, batch size $K = 40$), which indicates that we can conduct our attack successfully while a model is converging. The data indeed leaks to a certain level (PSNR above 20) while the model converges at a certain accuracy (0.68), which indicates that CAFE works in an attacking while training scenario.

### 4.4 Mitigation of CAFE data leakage attack via fake gradients

**Training and defense performance.** To demonstrate how fake gradients defend against CAFE (Section 3.4), we conduct CAFE with unchanged $\Theta$, which is the strongest data leakage attack setting. We use the SGD optimizer with learning rate set as 0.1, $\sigma^2 = 1.1$, and $\nu = 1000$ for fake gradients. Figure 6 shows a comparison between the visual image quality of the data recovered by CAFE on CIFAR-10 when the ordinary gradients and fake gradients are used, respectively. The PSNR of recovered data in CAFE on ordinary and fake gradients is 28.68 and 7.67, respectively. Moreover, Figure 7 shows that the training process with fakes gradients behaves in a similar way to the one with true gradients, confirming that the use of fake gradients does not lose the training efficacy. We have also added the experiment to discuss the difference of our fake gradients method to differential privacy (DP). The results and analysis are shown in Appendix G.3.

### 4.5 Recover human face data

We also implement CAFE on Yale $32 \times 32$ human face dataset [12], which achieves the PSNR above 42. The recovered data are shown in Appendix G.4. It implies that CAFE can fully recover data that requires privacy protection such as facial images.

## 5 Conclusions

In this paper, we uncover the risk of _catastrophic data leakage in vertical federated learning (CAFE)_ through a novel algorithm that can perform large-batch data leakage with high data recovery quality and theoretical guarantees. Extensive experimental results demonstrate that CAFE can recover large-scale private data from the shared aggregated gradients on vertical FL settings, overcoming the batch limitation problem in current data leakage attacks. We also propose an effective countermeasure using fake gradients to mitigate the potential risks of CAFE.

## Acknowledgments

This work was supported by National Science Foundation CAREER Award 2047177, and the Rensselaer-IBM AI Research Collaboration (http://airc.rpi.edu), part of the IBM AI Horizons Network (http://ibm.biz/AIHorizons). C-Y Hsu and C-M Yu were supported by MOST 110-2636-E-009-018, and we also thank National Center for High-performance Computing (NCHC) of National Applied Research Laboratories (NARLabs) in Taiwan for providing computational and storage resources.

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
