# OpenReview forum: "CAFE: Catastrophic Data Leakage in Vertical Federated Learning"
_NeurIPS.cc/2021/Conference — NeurIPS 2021 Poster_

### Official Review · Reviewer_PDMg · 2021-07-15

**Rating:** 7
**Confidence:** 3

**Summary:**

This paper studies recovering sensitive training data from vertical federated learning (VFL) process, where same set of training data is shared among the clients, but each has distinct set of features. The authors leveraged the data indices (or batch indices) to reliably recover internal representations, which in turn helps to recover the input training data with high quality. The experimental results validate the effectiveness of the proposed approach.

**Ethical Concerns:**

N/A.

**Limitations And Societal Impact:**

The authors addressed the limitations and potential negative impact.

**Main Review:**

$\textbf{Originality}$: I do not have enough knowledge about the related work, but utilizing the data indices to design stronger model inversion attack seems to be new. To the best of the reviewer's knowledge, related work is adequately cited.

$\textbf{Quality}$: the proposed approach is technically sound and the claims are well supported by the experimental results. I think it is a clever idea to utilize known data indices (although somewhat limits its application scenario) to recover some intermediate representations, which I believe gives the final optimization process (of recovering the input training data) an even finer control and improves the recovery quality.

$\textbf{Clarity}$: the paper is very well written and is easy to follow. There are some typos in the paper: in page 3, for the definition of $\mathcal{D}(s^t)$, it should be $s^t[i]$ instead of $s^t[n]$? "betch size" in page 7.

Regarding the baselines in page 8, the justifications for not including some methods in Table 1 is not satisfactory. For example, the reason "... require certain conditions which may fail to satisfy" is too ambiguous. I am fine with not including certain baselines with justifiable reasons, but they need to be very precise. When experimenting with the fake gradient based defense, it will be better to also report the training and testing accuracy for more intuitive understanding. In terms of writing, I think it will improve the paper if the authors can describe the high level motivation of designing multi-stage attack at the beginning of the paper. My current understanding is, multi-stage attack is preferred because it gives adversaries finer control over the optimization process of data recovery, and a side benefit is, some sensitive intermediate representations are also revealed. If my understanding is correct, then one follow-up question is, why not make the analysis even finer by recovering the inputs to the convolutional layers?

$\textbf{Significance}$: the method is not general enough and only applies to the special VFL setting where the data indices are known. In other FL settings, data indices are generally unavailable and the proposed method will not work. But I believe VFL is still an important setting and stronger attacks tailored for this specific setting are also valuable.

In the experiment section, comparison to the baselines might be an apple to orange comparison. The reviewer is only familiar with the two baselines [10], [26], but these two methods are actually very general and do not need the knowledge of data indices. Therefore, it is expected that in VFL setting with large batch sizes, this type of baselines will perform worse. I believe for the general FL setting where data indices are unavailable, larger batch size is still an effective defense.

When designing a defense in VFL setting, one straightforward option is to apply differential privacy during model training. I think the authors should discuss the difference of their method to differential privacy and it will be very impressive if the proposed method outperforms DP experimentally by providing better model utility while protecting the data privacy.

**Time Spent Reviewing:**

4

---

> ### Author Response · Authors · 2021-08-10
> **We thank Reviewer PDMg for detailed and constructive comments! (1/2)**
>
> We thank the reviewer’s support and the constructive comments. We address your remaining concerns below.
>
>
> **Q1. The training and testing accuracy of fake gradient based defense**
> We attached the training accuracy and testing accuracy when training with (Fake gradient) and without (True gradient) the proposed defense at $T=2500$ iterations based on the experiment setup of Figure 6 except for Linnaeus-5 with learning rate=0.01. From the table below, our defense strategy didn't affect model performance much. Note that one should compare the difference between the results using true v.s. fake gradients to study the effect of the defense. The low absolute value of the accuracy is an artifact of insuffieint training iterations due to time constraints.
>
> | Algorithm|Training loss| Training accuracy |Testing accuracy |
> | :-------------: | :----------: | :-----------: | :-----------: |
> |  True gradient (MNIST)  | 0.299  | 0.884  | 0.885  |
> |  Fake gradient (MNIST) | 0.404  | 0.857  | 0.858   |
> |  True gradient (CIFAR-10) | 2.198  | 0.263  | 0.263 |
> |  Fake gradient (CIFAR-10) | 2.236  | 0.247  | 0.248  |
> |  True gradient (Linnaeus-5) | 1.568  | 0.310  | 0.302 |
> |  Fake gradient (Linnaeus-5) | 1.652  | 0.292  | 0.279   |
>
>
> **Q2. Finer analysis on recovering the inputs to the convolutional layers**
>
> We apprecite the reviewer's insightful question. We have indeed tried to recover the inputs before a convolutional layer. Unfortunately, the attempt was not successful. Below we provide derivations to show why such recovery is impossible.
>
> In the simplest case, we assume a single input as $X \in \mathbb{R}^{M\times N\times C}$, the convolutional layer as $\Theta \in \mathbb{R}^{H\times W\times C \times G}$, the bias as $b \in \mathbb{R}^{G}$, the output as $Y \in \mathbb{R}^{I\times J \times G}$, the loss as $\mathcal{L} \in \mathbb{R}$, where $M$ and $N$ are the width and height of input $X$, $I$ and $J$ are the width and height of output $Y$，$C$ and $G$ are the number of input and output channels.
>
> We have
>
> \begin{align}
> 	Y_{i,j,g} = b_g + \sum_h\sum_w\sum_c \Theta_{h,w,c,g}X_{h+i, w+j,c}
> \end{align}
>
> and we can compute the gradient as
>
> \begin{align}
> 	\nabla_{\Theta_{h,w,c,g}}\mathcal{L} &= \frac{\partial\mathcal{L}}{\partial Y}\cdot\frac{\partial Y}{\partial \Theta_{h,w,c,g}}= \sum_i\sum_j \frac{\partial\mathcal{L}}{\partial Y_{i,j,g}}\cdot X_{h+i, w+j,c}.
> \end{align}
>
> Then we need to deal with the term $\frac{\partial\mathcal{L}}{\partial Y_{i,j,g}} \in \mathbb{R}^{I\times J\times R}$ for data recovery, and
> the gradient of bias is
>
> \begin{align}
> 	\nabla_{b_{g}}\mathcal{L} &= \frac{\partial\mathcal{L}}{\partial Y}\cdot\frac{\partial Y}{\partial b_{g}}= \sum_i\sum_j \frac{\partial\mathcal{L}}{\partial Y_{i,j,g}}.
> \end{align}
>
> In both equations of $\nabla_{\Theta_{h,w,c,g}}\mathcal{L}$ and $\nabla_{b_{g}}\mathcal{L}$, they contain the summation operations on both width and height dimension of output image $Y$. For each $i$ and $j$, $\frac{\partial\mathcal{L}}{\partial Y_{i,j,g}}$ is unknown to the server in standard VFL protocols. As the result, we cannot recover each element of $X$ from $\nabla_{\Theta_{h,w,c,g}}\mathcal{L}$ and $\nabla_{b_{g}}\mathcal{L}$ even when the batch size is $1$. Furthermore, there usually exist max or average pooling layers in convolutional neural network structures, making it more difficult to recover the input $X$. However, in fully connected (FC) layers, there is no summation operations when we compute the gradients of loss w.r.t bias of an FC layer (equation (8) and (9) in [Geiping et al., 2020]), where we can directly recover the input $H$ to an FC layer. We will add this discussion to the revised version.
>
> [Ref [10]] J. Geiping, H. Bauermeister, H. Dröge, and M. Moeller, "Inverting gradients – how easy is it to break privacy in federated learning?" *NeurIPS 2020*
>
> **Q3. The method is not general enough and only applies to the special VFL setting.**
>
> We understand the reviewer's concern; but we disagree our attack only applies to "special" VFL setting. There are many practical industrial examples of VFL that fit into our considered setting, such as finance, e-commerce, and health. For example, vertical federated learning (VFL) provides data indices for CAFE's implementation, and it is an important FL paradigm that is suitable in cases where multi-local datasets or data owners share the same data identity (ID) space but their data differ in feature space. There are many practical industrial examples of VFL such as
> + [FedML at Webank AI](https://github.com/FedML-AI/FedML/tree/master/fedml_experiments/standalone/classical_vertical_fl) [C.He et al., 2020]
> + [FATE at Webank AI](https://github.com/FederatedAI/FATE)
> + [TF-Encrypted by Alibaba Gemini Lab](https://alibaba-gemini-lab.github.io/docs/blog/tfe/)
>
> Moreover, we can find the similar VFL protocol in other papers such as
> + Haiqin Weng, Juntao Zhang, Feng Xue, Tao Wei, Shouling Ji, Zhiyuan Zong "Privacy Leakage of Real-World Vertical Federated Learning", *arXiv, eprint:2011.09290*
> + Cheng, Kewei and Fan, Tao and Jin, Yilun and Liu, Yang and Chen, Tianjian and Papadopoulos, Dimitrios and Yang, Qiang "SecureBoost: A Lossless Federated Learning Framework", *IEEE Intelligent Systems, 2021*
> + Fanglan Zheng, Erihe, Kun Li, Jiang Tian, Xiaojia Xiang "A Vertical Federated Learning Method for Interpretable Scorecard and Its Application in Credit Scoring", *arXiv, eprint:2009.06218*
> + Stephen Hardy, Wilko Henecka, Hamish Ivey-Law, Richard Nock, Giorgio Patrini, Guillaume Smith, Brian Thorne "Private federated learning on vertically partitioned data via entity resolution and additively homomorphic encryption", *arXiv, eprint:1711.10677*
> + Chaoyang He, Songze Li, Jinhyun So, Xiao Zeng, Mi Zhang, Hongyi Wang, Xiaoyang Wang, Praneeth Vepakomma, Abhishek Singh, Hang Qiu, Xinghua Zhu, Jianzong Wang, Li Shen, Peilin Zhao, Yan Kang, Yang Liu, Ramesh Raskar, Qiang Yang, Murali Annavaram, Salman Avestimehr "FedML: A Research Library and Benchmark for Federated Machine Learning" *NeurIPS 2020*
>
>
> We will change the title and descriptions in the article from FL to VFL to avoid unnecessary confusion.
>
> **Q4. Issue with the baseline comparison**
>
> Following your suggestion, we expand the reasons to why some methods summarized in Table 1 are not compared to our proposed CAFE method.
>
> Theory-driven label inference methods are proposed in [B. Zhao et al., 2020] and in [W. Wei et al., 2020], and the authors proposed label-based regularizers. However, our attack mainly deals with training data leakage, so we didn't compare our work with training label inference methods. In [X. Pan et al., 2020], the authors proposed a sufficient requirement that "each data sample has at least two exclusively activated neurons at the last but one layer". However, in our training protocol, the batch size is too large and it is almost impossible to ensure that each selected data sample has at least two exclusively activated neurons. In [J. Zhu et al. 2020], the authors claim that in the batch setting, their method will only return a linear combination of the selected training data, which is a very restricted assumption.
>
> In GradInversion [Yin et al., 2021], the authors proposed several assumptions such as non-repeating labels in the batch, which is hard to be satiesfied in some datasets we used like CIFAR-10, MNIST and Linnaeus-5. In those datasets, we use batch size of more than $40$, which is larger than the number of classes ($10$ or $5$), and therefore the assumption in the paper does not hold.  Nevertheless, during the limited rebuttal period, we still compared our CAFE to the methods by using the batch normalization regularizer (eq. (10) in [Yin et al., 2021]) and group consistency regularizer (eq. (11) in [Yin et al., 2021]) mentioned in their work in CAFE. We report the results below using the same experiment setup in Table 2. The results in the table below show that CAFE outperforms these two methods by a large margin. We will add these results in the revised version.
>
>
> | Baselines\PSNR\Datasets|CIFAR-10| MNIST |Linnaeus-5|
> | :------------- | :----------: | :-----------: | :-----------: |
> |  CAFE | 31.83   | 43.15    | 33.22 |
> | BN regularizer [Yin et al., 2021]   | 18.94 | 13.68 | 8.09 |
> | Group consistency regularizer [Yin et al., 2021]   | 13.63 | 9.24 | 12.32 |
>
> If the reviewer has any specific baseline in mind, please let us know and we are willing to compare and report the results.
>
> [Ref [21]] X. Pan, M. Zhang, Y. Yan, J. Zhu, and M. Yang, “Theory-oriented deep leakage from gradients via linear equation solver”, *ArXiv eprint:2010.13356 2020*
>
> [Ref [24]] W. Wei, L. Liu, M. Loper, K. H. Chow, M. E. Gursoy, S. Truex, and Y. Wu, “A framework for evaluating gradient leakage attacks in federated learning”, *ArXiv eprint:2004.10397 2020*
>
> [Ref [27]] Hongxu Yin, Arun Mallya, Arash Vahdat, Jose M. Alvarez, Jan Kautz, Pavlo Molchanov "See through Gradients: Image Batch Recovery via GradInversion" *CVPR 2021*
>
> [Ref [28]] B. Zhao, K. R. Mopuri, and H. Bilen, “idlg:  Improved deep leakage from gradients”, *arXiv,eprint:2001.02610, 2020*
>
> [Ref [29]] J. Zhu and M. B. Blaschko, “R-GAP: Recursive gradient attack on privacy”, *in International Conference on Learning Representations, 2021*

---

> > ### Author Response · Authors · 2021-08-10
> > **We thank Reviewer PDMg for detailed and constructive comments! (2/2)**
> >
> > **Q5. Discussion on the difference of the defense method to the differential privacy**
> >
> > Following the reviewer's suggestion, we have also added the experiment to discuss the difference of our fake gradient method to differential privacy.
> >
> > The results below show the training loss of no defense (true gradient), differential privacy (DP) defense, and our defense (fake gradient). For DP, we followed the gradient clipping approach [M. Abadi et al. *ACM CCS, 2016*] to apply DP to the gradients from workers. In particular, the gradient norm was clipped to 3, as suggested by [M. Abadi et al. *ACM CCS, 2016*]. As shown in the table below, the training loss cannot be effectively reduced using DP. This is also consistent with the result in [Zhu et al. *NeurIPS, 2019*], as one proposed to add noise to gradient as a candidate defense. [Zhu et al. *NeurIPS, 2019*] concluded that the noise with sufficient magnitude required to avoid information leakage from gradients degrades the accuracy significantly. As the noise magnitude required by DP is even stronger than the one needed for the ad hoc privacy in [Zhu et al. *NeurIPS, 2019*], it is inevitable to lead to a similar conclusion, which is the difficulty in reducing the training loss. In our fake gradient defense, all of the gradients will be projected to a set of predefined gradients before being sent to the server, with the purpose of restricting the attacker’s knowledge from gradient leakage. Our defense is still deterministic in its essence and therefore does not satisfy the DP. However, as DP may largely destabilize the training process, our defense strikes a balance between the empirical data privacy and model accuracy. In other words, our experiments demonstrate that the attacker is unable to recover the worker’s data and at the same time the training loss can be reduced effectively. We will add this discussion in the revised version.
> >
> >
> > |  \# of iterations ($T$) \ Training loss \ DP|DP $\epsilon=10$|DP $\epsilon=5$|DP $\epsilon=1$|DP $\epsilon=0.1$|Fake gradients|True gradients|
> > | :------------- | :----------: | :-----------: | :-----------: | :-----------: | :-----------: | :-----------: |
> > |  0  | 2.78  | 2.77  | 2.77  | 2.77  | 2.77  | 2.77  |
> > |  1K | 2.69  | 2.69  | 2.69  | 2.69  | 1.95  | 1.08  |
> > |  2K | 2.85  | 2.85  | 2.85  | 2.85  | 1.38  | 0.54  |
> > |  3K | 2.85  | 2.85  | 2.85  | 2.85  | 0.65  | 0.23  |
> > |  4K | 2.92  | 2.92  | 2.92  | 2.92  | 1.09  | 0.38  |
> > |  5K | 2.61  | 2.61  | 2.61  | 2.61  | 0.92  | 0.31  |
> > |  6K | 2.69  | 2.69  | 2.69  | 2.69  | 0.62  | 0.31  |
> > |  7K | 2.77  | 2.77  | 2.77  | 2.77  | 1.08  | 0.15  |
> > |  8K | 2.69  | 2.69  | 2.69  | 2.69  | 1.15  | 0.46  |
> >
> > [Ref [30]] Zhu et al. "Deep Leakage from Gradients"* NeurIPS, 2019*
> > M. Abadi et al. "Deep Learning with Differential Privacy", *ACM CCS, 2016*
> >
> > **Q6. Some typos**
> >
> > Thank you for the careful reading. We will correct all the typos that you have pointed out in our next version.

---

### Official Review · Reviewer_a7yr · 2021-07-15

**Rating:** 6
**Confidence:** 4

**Summary:**

This paper proposes a method for the server to recover the batch data from the gradients obtained in vertical federated learning. The recovery process is based on the use of data indices of the batch which are known.  The recovery contains three steps, and the most important nouvel part is to recover the first input of a fully connected layer for every data sample in the batch ($H$ in the paper) and then introduce it as a metric in the final recovery objective function.

**Limitations And Societal Impact:**

Yes, the authors have adequately addressed the limitations and potential negative societal impact of their work

**Main Review:**

CAFE recovers the whole data as it makes use of the indices of the batch that other previous strategies have no access to. It keeps recovering every image as time goes on i.e., making use of all the collected gradients. If the global model is unchanged (not in a training mode), the authors prove that CAFE recovers the optimal $H$.

Overall the strategy is quite original but with strong assumptions. It may be better to mention in the title that it is for vertical federated learning. As for horizontal federated learning (the one we think about first when we mention FL), it is not realistic that the server would know the batch indices.  Besides, as the authors pointed out, the attack will be weak while training (that’s to say that $\Theta$ changes according to the FL procedure), as it is harder to recover $H$ which depends on $\Theta$. Thus, CAFE works well only when the learning rate in training is small. This limitation is better to be mentioned in the early pages. Although the leakage seems to be catastrophic, there are limitations for the efficiency of this attack.

Some other comments:
1. Lines 221 - 224: Can you explain why data leakage performance of CAFE is independent of the batch size? Do the experiments in Table 3 share the same T? If the same T is used, from the attack perspective, larger batch size may help the algorithm to approximate $H$ and thus favors the CAFE performance.
2. I got confused about the names of the algorithm 3 and 4. Should algorithm 3 be the single-loop one and algorithm 4 be the nested one? There are contradictory conclusions on the performance of these algorithms in lines 167-168 and lines 240-241.
3. It may be interesting to show the convergence plot for experiment 4.3 with small learning rate. In addition, as mentioned in the paragraph, it would be great to study as well if CAFE works better on an untrained model.
4. In algorithm 4, line 5 should be “Run step 11-13” and line 6 should be “Run step 11-13”?



**Time Spent Reviewing:**

10

---

> ### Author Response · Authors · 2021-08-10
> **We thank Reviewer a7yr for detailed and constructive comments!**
>
> First of all, we thank the reviewer for the constructive comments and careful reading. We will address your concerns below.
>
> **Q1. Regarding the title and the scope.**
>
> Following your suggestion, we will change the title and descriptions in the article from FL to VFL, which is a more precise statement. In Section 3 we have clearly mentioned that we focus on VFL setting. But we agree we should make our scope clearer by specifying VFL upfront to avoid unnecessary confusion. We will also follow your suggestion to mention CAFE is less effective when the learning rate is large in earlier pages.
>
>
> **Q2. The term $T$ in our experiment.**
>
> Theoretically, given infinite number of iterations, we prove that we can recover $\nabla_{U}\mathcal{L}$ and $H$ because the respective objective function in equation (7) and (9) in our paper is strongly convex as long as $N < d_2$ and $\text{Rank}(\textbf{V}^{*})=N$ (see details in the supplementary material $D$ and $E$).
>
> In our experiment, we did fix the number $T$ for each dataset and it shows that large batch size indeed helps the CAFE algorithm to approximate $H$, especially in MNIST. Following the reviewer's comment, we also conducted an experiment using the same number of epochs on Linnaeus-5 (same setup in Table 3) and reported the results in the table below. The results suggest that increasing batch size $K$ (Table 3 in the paper) and number of iterations $T$ (Table below) both contribute to the attack performnace. When we fix the number of epoches, the attacker with a smaller batch size has more iterations to recover data, leading to a better performance. We will add this result in the revised version.
>
> | **Epoch**\PSNR\Batch size| $10$|$20$|$40$|$80$|$100$|
> | :------------- | :----------: | :-----------: | :-----------: | :-----------: | :-----------: |
> |   100 | 12.30 | 14.76 | 15.33 | 11.84 | 11.79 |
> |  150 | 15.83 | 17.92 | 16.26 | 14.28 | 13.21 |
> |  200 | 17.63 | 19.38 | 17.20 | 16.24 | 14.46 |
> |  250 | 21.80 | 21.49 | 19.09 | 18.11 | 16.14 |
> |  300 | 22.92 | 24.00 | 21.14 | 19.83 | 17.29 |
> |  350 | 24.86 | 25.86 | 22.62 | 21.05 | 18.90 |
>
> **Q3. Typos in line 240-243 and Algorithm 3, 4**
>
> Thank you for correcting the typos. It is indeed a typo in line 240-243 and it should be "As shown in Table 5, CAFE single-loop requires fewer number of iterations. Meanwhile, it is difficult to set the loop stopping criteria in the CAFE nested-loop mode."
>
> In addition, line 5 in Algorithm 4 should be “Run step 11-13” and line 6 should be “Run step 11-13”. We apologize for our typos again.
>
> **Q4. It may be interesting to show the convergence plot for experiment 4.3 with small learning rate. Is CAFE better on untrained model?**
>
> Following the reviewer's suggestion, we have conducted a new experiment on MNIST with $4$ workers. The model indeed converges with a relative small learning rate (we used Adam optimizer, learning rate = $10^{-6}$, trained on $800$ images, tested on $100$ images, batch size $K=40$) and we attach our experimental results below.
>
> | \# of iterations ($T$)  |PSNR value|Training loss|Testing accuracy|
> | :------------- | :----------: | :-----------: | :-----------: |
> |   0  | 5.07 | 2.36 | 0.11 |
> |  2K  | 11.68 | 2.31 | 0.27 |
> |  4K  | 14.39 | 2.14 | 0.34 |
> |  6K  | 18.07 | 1.99 | 0.54 |
> |  8K  | 18.16 | 1.87 | 0.63 |
> |  10K | 18.12 | 1.82 | 0.64 |
> |  15K | 16.86 | 1.63 | 0.65 |
> |  20K | 20.72 | 1.68 | 0.68 |
>
> The data indeed leaks to a certain level (PSNR above 20) while the model converges at a certain testing accuracy ($0.68$), which indicates that CAFE works in an attacking while training scenario. We note that due to the stealthy nature of CAFE, it did not impede the performance of the final model. The relatively low accuracy is merely because only $800$ samples are used for training.
>
>
> CAFE is indeed better to recover data when the model is untrained comparing with the stage when the model  fully converges. We use our theoretical analysis to explain the reason. When the model is trained or even fully converged, the real gradients of loss with repect to (w.r.t) model parameters will be very small. It is possible that the value of the recoverd $\nabla_{U}\mathcal{L}$ in equation (7) will also be close to 0. In that case, it can be difficult to recover $H$ in equation (9).

---

### Official Review · Reviewer_kAc7 · 2021-07-16

**Rating:** 5
**Confidence:** 3

**Summary:**


This paper proposes an information leakage attack (i.e., CAFE) that can recover data samples from aggregated gradients against a  FL setting. The proposed attack first recovers gradients of loss with respect to the outputs of the first fully connected (FC) layer. It then recovers the first FC layer's inputs. Finally, it recovers the original data samples by optimizing an objective that consists of the traditional gradient matching objective and two auxiliary regularizers:  a TV norm term and an internal representation regularization term. The paper provides both theoretical analysis and numerical verification about CAFE's performance. Experiments show that the proposed method outperforms some of the previous works.

**Limitations And Societal Impact:**

See the main review.

**Main Review:**

[Strengths]

1. The paper presents an information leakage attack against FL.
2. In addition to numerical results, the paper also presents a theoretical guarantee of the proposed method's performance.
3. The paper also provides a defense mechanism to mitigate the proposed attack.

[Weaknesses]

1. My main concern of the proposed method is that it might not generalize to a typical FL setting where the clients only send their own model updates to the server (without communicating with other workers). The authors might want to clarify how the proposed method can generalize to the traditional FL setting. Otherwise, the contribution of the paper would be quite limited.
2. In the experiments, only three baselines are selected: DLG, cosine similarity, and SAPAG. More recent and more advanced methods  (e.g., Inverting gradients [Geiping et al., 2020], GradInversion [Yin et al., 2021]) are not included for comparison.
3. The FL setting in the experiments only includes four workers. In practice, an FL system usually consists of hundreds of or even thousands of clients. The proposed method works in VFL, in which local agents need to exchange updates with other workers to compute the gradients.
4. Some of the concepts are a bit vague. For instance, the paper mentions "data index alignment." However, it does not provide a clear definition or explanation.

**Time Spent Reviewing:**

12

---

> ### Author Response · Authors · 2021-08-10
> **We thank Reviewer kAc7 for detailed and constructive comments!**
>
> First of all, we thank the reviewer for the careful review and constructive feedback. Below we address your concerns.
>
> **Q1. The considered federated learning protocol is not typical.**
>
>
> We would like to clarify that our proposed CAFE framework actually applies to the described typical federated learning (FL) setting. Because CAFE is a server-end attack, it is agnostic to the assumption of inter-worker communication (or not), as long as data indices are aligned (see Line 121-124).
>
> For example, many vertical federated learning (VFL) protocols provide data indices, and VFL is an important FL paradigm which is suitable in cases where multi local datasets or data owners share the same data identity (ID) space but their data differ in feature space. Some industrial examples of VFL are:
> + [FedML at Webank AI](https://github.com/FedML-AI/FedML/tree/master/fedml_experiments/standalone/classical_vertical_fl) [C.He et al., 2020]
> + [FATE at Webank AI](https://github.com/FederatedAI/FATE)
> + [TF-Encrypted by Alibaba Gemini Lab](https://alibaba-gemini-lab.github.io/docs/blog/tfe/)
>
> Moreover, we can find similar VFL protocols that fit into our studied scenario in other papers such as
> + Haiqin Weng, Juntao Zhang, Feng Xue, Tao Wei, Shouling Ji, Zhiyuan Zong "Privacy Leakage of Real-World Vertical Federated Learning", *arXiv, eprint:2011.09290*
> + Cheng, Kewei and Fan, Tao and Jin, Yilun and Liu, Yang and Chen, Tianjian and Papadopoulos, Dimitrios and Yang, Qiang "SecureBoost: A Lossless Federated Learning Framework", *IEEE Intelligent Systems, 2021*
> + Fanglan Zheng, Erihe, Kun Li, Jiang Tian, Xiaojia Xiang "A Vertical Federated Learning Method for Interpretable Scorecard and Its Application in Credit Scoring", *arXiv, eprint:2009.06218*
> + Stephen Hardy, Wilko Henecka, Hamish Ivey-Law, Richard Nock, Giorgio Patrini, Guillaume Smith, Brian Thorne "Private federated learning on vertically partitioned data via entity resolution and additively homomorphic encryption", *arXiv, eprint:1711.10677*
> + Chaoyang He, Songze Li, Jinhyun So, Xiao Zeng, Mi Zhang, Hongyi Wang, Xiaoyang Wang, Praneeth Vepakomma, Abhishek Singh, Hang Qiu, Xinghua Zhu, Jianzong Wang, Li Shen, Peilin Zhao, Yan Kang, Yang Liu, Ramesh Raskar, Qiang Yang, Murali Annavaram, Salman Avestimehr "FedML: A Research Library and Benchmark for Federated Machine Learning" *NeurIPS 2020*
>
>  Therefore, our considered FL setting is typical and our contributions are not limited.
>
> **Q2. Insufficient reference to more advanced and recent papers such as e.g., Inverting gradients [Geiping et al., 2020], GradInversion [Yin et al., 2021].**
>
> In fact, we have already compared to the requested baseline [Geiping et al., 2020] and discussed these two mentioned works in Table $2$. In our paper we call the baseline [Geiping et al., 2020] "cosine similarity'' since the authors didn't give a name of their algorithm.
>
> In GradInversion [Yin et al., 2021], the authors proposed several assumptions such as non-repeating labels in the batch, which is hard to be satiesfied in some datasets we used like CIFAR-10, MNIST and Linnaeus-5. In those datasets, we use batch size of more than $40$, which is larger than the number of classes ($10$ or $5$), and therefore the assumption in the paper does not hold.  Nevertheless, during the limited rebuttal period, we still compared our CAFE to the methods by using the batch normalization regularizer (eq. (10) in [Yin et al., 2021]) and group consistency regularizer (eq. (11) in [Yin et al., 2021]) mentioned in their work in CAFE. We report the results below using the same experiment setup in Table 2. The results show that CAFE outperforms these alternatives. We will add these results in the revised version.
>
>
> | Baselines\PSNR\Datasets|CIFAR-10| MNIST |Linnaeus-5|
> | :------------- | :----------: | :-----------: | :-----------: |
> |  CAFE | 31.83   | 43.15    | 33.22 |
> | BN regularizer [Yin et al., 2021]   | 18.94 | 13.68 | 8.09 |
> | Group consistency regularizer [Yin et al., 2021]   | 13.63 | 9.24 | 12.32 |
>
> [Ref [10]] J. Geiping, H. Bauermeister, H. Dröge, and M. Moeller, "Inverting gradients – how easy is it to break privacy in federated learning?" *NeurIPS 2020*
>
> [Ref [27]] Hongxu Yin, Arun Mallya, Arash Vahdat, Jose M. Alvarez, Jan Kautz, Pavlo Molchanov "See through Gradients: Image Batch Recovery via GradInversion" *CVPR 2021*
>
>
> **Q3. The number of local workers is not sufficient.**
>
> We explain the effect of the number of local workers on CAFE in the VFL case from both theoretical and practical aspects.
>
> In the paper, we define the input data $\textbf{x}_n$ in line $87$-$88$. Although the data is partitioned on feature space by local workers, the dimension of the entire data feature space is fixed and indepedent of the number of workers. Therefore, increasing number of workers theoretically does not change the dimension of variables assosicated with data recovery in equation (3) in our paper.
>
> In practice, different from HFL, where there could be hundreds of workers, in VFL, the workers are typically financial organizations or companies. Therefore, the number of workers is usually small (See FedML at Webank AI as an example).
>
> In the table below we added a new experiment with 16 workers following the same experiment setup as in Table 2. The results show that the CAFE performances are comparable.
>
>
> | Baselines\PSNR\Datasets|CIFAR-10| MNIST |Linnaeus-5|
> | :------------- | :----------: | :-----------: | :-----------: |
> |  CAFE w/ $4$ workers | 31.83   | 43.15    | 33.22 |
> | CAFE w/ $16$ workers   | 28.39 | 39.28 | 39.85 |
>
>
> **Q4. Definition of data index alignment.**
>
> We are sorry for the confusion. We now realize we discussed this term in Line 121-124 but did not define it formally. The term "data index alignment" means that for each batch, the server (acting as the attacker) sends a data index or data id list to all the local workers to ensure that data with the same id sequence have been selected by each worker, which is a typical setting in VFL. Fig 2(b) in [Yang et al ACM 2019] gives an appropriate illustration of this step. We will clarify this concept in detail in the next version.
>
> [Ref [26]] Q. Yang, Y. Liu, T. Chen, and Y. Tong, “Federated machine learning: Concept and applications” *ACM Trans. Intell. Syst. Technol., vol. 10, no. 2, Jan. 2019*

---

> > ### Author Response · Authors · 2021-08-27
> > **Does our response address your concerns?**
> >
> > Since the rolling discussion phase is closing soon, we would like to make sure your concerns have been fully addressed by our response. In particular, we have clarified why our considered vertical federated learning is standard and added two additional experiments:
> > 1. To compare CAFE with the related works you suggested;
> > 2. To evaluate the data leakage performance of CAFE with more workers.
> >
> > If there is anything we can do to convince you about the merits of this work, please let us know!

---

### Author Response · Authors · 2021-08-10
**General Response**

**[Scope]** We appreciate the constuctive reviews from all reviewers. The reviewers had a common comment that our contributions are catestrophic data leakage in vertical federated learning (VFL) with theoretical analysis and emprirical results, but the scope is not clearly reflected in our title and early sections. We will make the suggested title changes and revise the early sections for clarity.

**[VFL]** We would also like to emphasize that our considered FL setting is typical and there are many practical industrial examples of VFL such as finance, e-commerce, and health.
For example, many VFL protocols provide data indices, and VFL is an important FL paradigm which is suitable in cases where multi local datasets or data owners share the same data identity (ID) space but their data differ in feature space. Some industrial examples of VFL are:
+ [FedML at Webank AI](https://github.com/FedML-AI/FedML/tree/master/fedml_experiments/standalone/classical_vertical_fl) [C.He et al., 2020]
+ [FATE at Webank AI](https://github.com/FederatedAI/FATE)
+ [TF-Encrypted by Alibaba Gemini Lab](https://alibaba-gemini-lab.github.io/docs/blog/tfe/)

Moreover, we can find similar VFL protocols that fit into our studied scenario in other papers such as

+ Cheng, Kewei and Fan, Tao and Jin, Yilun and Liu, Yang and Chen, Tianjian and Papadopoulos, Dimitrios and Yang, Qiang "SecureBoost: A Lossless Federated Learning Framework", *IEEE Intelligent Systems, 2021*
+ Haiqin Weng, Juntao Zhang, Feng Xue, Tao Wei, Shouling Ji, Zhiyuan Zong "Privacy Leakage of Real-World Vertical Federated Learning", *arXiv, eprint:2011.09290*
+ Fanglan Zheng, Erihe, Kun Li, Jiang Tian, Xiaojia Xiang "A Vertical Federated Learning Method for Interpretable Scorecard and Its Application in Credit Scoring", *arXiv, eprint:2009.06218*
+ Stephen Hardy, Wilko Henecka, Hamish Ivey-Law, Richard Nock, Giorgio Patrini, Guillaume Smith, Brian Thorne "Private federated learning on vertically partitioned data via entity resolution and additively homomorphic encryption", *arXiv, eprint:1711.10677*
+ Chaoyang He, Songze Li, Jinhyun So, Xiao Zeng, Mi Zhang, Hongyi Wang, Xiaoyang Wang, Praneeth Vepakomma, Abhishek Singh, Hang Qiu, Xinghua Zhu, Jianzong Wang, Li Shen, Peilin Zhao, Yan Kang, Yang Liu, Ramesh Raskar, Qiang Yang, Murali Annavaram, Salman Avestimehr "FedML: A Research Library and Benchmark for Federated Machine Learning" *NeurIPS 2020*

Below we summarize the key points in our response to each reviewer. The detailed point-by-point response can be found underneath each reviewer's review.

**Reviewer kAc7**
* Added more references and real-world examples of VFL on Q1 (The learning protocol is not typical)
* Added experimental baseline comparison results to our CAFE on Q2 (Insufficient baseline comparison)
* Added experimental results comparing different number of workers on Q3 (Insufficient number of workers)
* Added more clear descriptions on Q4 (Unclear concepts)

**Reviewer a7yr**
* Revised some statement on Q1 (Regarding the title and the scope)
* Added both discussion and experimental results on batch size $K$ and number of iterations $T$ on Q2 (The term $T$ in our experiment)
* Revised some typos on Q3 (Some typos)
* Added discussion and experimental results on the attacking while training case on Q4 (Convergence plot with a small learning rate)

**Reviewer PDMg**
* Added experimental training and testing accuracy of fake gradient based defense on Q1 (training and testing accuracy of fake gradient based defense)
* Added theoretical analysis on input recovery to a convolutional layer on Q2 (Finer analysis on recovering the inputs to the convolutional layers)
* Added more references and real-world examples of VFL on Q3 (Not general enough FL setting)
* Added experimental baseline comparison results to our CAFE on Q4 (Issue with the baseline comparision)
* Added both experimental differential privacy (DP) results and discussions on Q5 (The difference of the method to differential privacy)
* Revised some typos on Q6 (Some typos)


We would like to make the most use of the interactive discussion function provided by Openreview to clarify any concern the reviewers may have. We look forward to the rolling dicussion and further engagement with the reviewers and area chair(s)!

---

> ### Public Comment · ~Cheng_Hong1 · 2022-08-10
> **TF-Encrypted is not a FL framework but an MPC framework**
>
> First, congratulations on your NeurIPS paper acceptance, it is one of the few good papers that focus on the security issues of Federated learning.
>
> I am one of the contributors of TF-Encrypted, and was directed to this post by some colleagues.  Let me make a clarification:
> TF-Encrypted is not a FL framework but a Secure Multi-party Computation (MPC) framework, where all the intermediate information exchanged between participants are secret-shared, i.e. indistinguishable with random numbers. There're no such things like "aggregated gradients" in MPC, they only exist in FL.
>
>
>
> Thanks.
>
> Cheng Hong,
>
> Gemini Lab, Alibaba Group

---

> > ### Public Comment · Authors · 2022-08-11
> > **Clarification for TF-Encrypted**
> >
> > Dear Cheng,
> >
> > Thanks a lot for your kind words and your interest in our work! Sorry for the imprecise wording. Indeed, we agree that TF-Encrypted is more an MPC framework rather than an FL framework.
> >
> > Yours,
> >
> > Authors

---

### Author Response · Authors · 2021-08-23
**Looking forward to reviewers' feedback**

Dear reviewers,

We would like to start by thanking all reviewers for the positive feedback and constructive comments given in the initial reviews. While the discussion deadline is approaching, we have not received any feedback based on our responses. We would like to use the interactive feature of OpenReview to engage the reviewers with the discussion. In particular, in our responses, we believe we have provided additional clarifications and new numerical results to fully address the reviewer's concerns. We hope our responses convince the reviewers about the merits of this work. If the reviewer has any other suggestions or comments, please don't hesitate to let us know!

Best Regards,

Authors

---

### Comment · Reviewer_PDMg · 2021-08-24
**Thanks for the response on raised concerns**

I would like to thank the authors for providing point-to-point response on my raised concerns. My concerns are mostly addressed by the author rebuttal. Thanks!

---

> ### Author Response · Authors · 2021-08-24
> **Thank you for your prompt response**
>
> We thank the reviewer for your prompt response. We are delighted to learn that your concerns are mostly addressed by our response.

---

### Decision · Program_Chairs · 2021-09-27

**Decision:**

Accept (Poster)

**Comment:**

The reviewers were not strongly positive about the paper. The reviewers had a recurring concern over the distinction between Vertical Federated Learning (VFL) and Horizontal Federated Learning, probably the more common one. I would recommend the authors to update the title and make a clear distinction in the introduction itself, to keep the expectation of the reader focussed on VFL. Also, reviewer kAc7 had some concerns about the vagueness in the definition of concepts in the paper, which needs to be addressed.